# Old and New Beta-Lactamase Inhibitors: Molecular Structure, Mechanism of Action, and Clinical Use

**DOI:** 10.3390/antibiotics10080995

**Published:** 2021-08-17

**Authors:** Davide Carcione, Claudia Siracusa, Adela Sulejmani, Valerio Leoni, Jari Intra

**Affiliations:** 1Department of Laboratory Medicine, University of Milano-Bicocca, Azienda Socio Sanitaria Territoriale Brianza ASST-Brianza, Desio Hospital, Via Mazzini 1, 20833 Desio, Italy; davide.carcione@asst-brianza.it (D.C.); claudia.siracusa@asst-brianza.it (C.S.); a.sulejmani@campus.unimib.it (A.S.); 2Clinical Chemistry Laboratory, University of Milano-Bicocca, Azienda Socio Sanitaria Territoriale di Monza ASST-Monza, San Gerardo Hospital, Via Pergolesi 33, 20900 Monza, Italy; j.intra@asst-monza.it

**Keywords:** antibiotic resistance, β-lactamase inhibitor, β-lactams, carbapenem, penicillin, cephalosporin

## Abstract

The β-lactams have a central place in the antibacterial armamentarium, but the increasing resistance to these drugs, especially among Gram-negative bacteria, is becoming one of the major threats to public health worldwide. Treatment options are limited, and only a small number of novel antibiotics are in development. However, one of the responses to this threat is the combination of β-lactam antibiotics with β-lactamase inhibitors, which are successfully used in the clinic for overcoming resistance by inhibiting β-lactamases. The existing inhibitors inactivate most of class A and C serine β-lactamases, but several of class D and B (metallo-β-lactamase) are resistant. The present review provides the status and knowledge concerning current β-lactamase inhibitors and an update on research efforts to identify and develop new and more efficient β-lactamase inhibitors.

## 1. Introduction

The β-lactams are the most common antibiotics prescribed and used in a large variety of infectious diseases [1,2]. Several penicillin derivatives and related β-lactam classes, such as cephalosporins, cephamycins, monobactams, and carbapenems, have been discovered in the last century. Each new antibiotic was developed to increase the number of bacterial species covered by antimicrobial activity and to bypass resistance mechanisms arising from microbes, such as drug inactivation or modification by β-lactamases, alteration of binding sites, alteration of metabolic pathways, and decreased permeability or increased efflux across the membrane. Beta-lactamases are bacterial enzymes that hydrolyze the β-lactam bonds in β-lactam antibiotics, and they are divided into four classes according to the primary sequence homology and the differences in hydrolytic mechanisms: A, B, C, and D [2,3,4,5]. Classes A, C, and D β-lactamases are serine enzymes, which can hydrolyze the β-lactam ring via a serin-bound acyl intermediate in the active site, whereas class B β-lactamases (named metallo-β-lactamases, MBLs) present one or two zinc ions in the active site, which are necessary for their enzymatic activity [2,5]. However, the continued use of β-lactams, which is often excessive and inappropriate, has led to the spread of resistance to β-lactams, to extended-spectrum cephalosporins (e.g., cefotaxime, ceftriaxone, and ceftazidime), and more recently to carbapenems (doripenem, ertapenem, imipenem, meropenem). In fact, a worldwide health public problem is the proliferation of carbapenem-resistant bacteria due to carbapenem-hydrolyzing β-lactamases, such as *Klebsiella pneumoniae* carbapenemase (KPC) belonging to class A, New Delhi metallo-β-lactamase (NDM) belonging to class B, and oxacillinase (OXA-48) belonging to class D [2,5]. The global effort to prevent β-lactam resistance is the development of broad-spectrum β-lactamase inhibitors which mimic the β-lactam core, thus blocking β-lactamases, also including cephalosporinases and serine-based carbapenemases, that severely limit the antimicrobial activity [1,2,5]. The most common bacteria treated by β-lactamase inhibitors are the Enterobacteriaceae (*Escherichia coli*, *Klebsiella pneumoniae*, *Proteus vulgaris*, *Citrobacter* spp., *Salmonella* spp., *Shigella*), *Acinetobacter baumannii*, *Haemophilus influenzae*, *Mycobacterium tuberculosis*, *Neisseria gonorrhoeae*, and *Pseudomonas aeruginosa* [1,2,5]. The first combinations of β-lactam/β-lactam inhibitors that have been used over a long period are amoxicillin/clavulanic acid, ampicillin/sulbactam, and piperacillin/tazobactam. New approaches aimed to develop and to expand the number and effectiveness of β-lactamase inhibitors are in progress [5,6]. The purpose of this work is to provide a comprehensive overview of β-lactamase inhibitors currently used, as well as information concerning new compounds that have recently been developed or that are in progress.

## 2. A Global Overview of Genes Encoding β-Lactamases

Several Gram-negative bacteria possess naturally occurring, chromosomally mediated β-lactamases that were described as helping the microorganisms to compete with β-lactam-producing bacteria or to remove β-lactam-like molecules that can be used as regulators of cell wall synthesis. At least 400 different types of β-lactamase have been reported. The most common plasmid mediated class A β-lactamases in Gram-negative bacteria, including Enterobacteriaceae*, P. aeruginosa, H. influenzae*, and *N. gonorrhoeae*, were TEM and a closely related enzyme named TEM-2. These two enzymes can hydrolyze penicillins and a few cephalosporins. In fact, they are not effective against third-generation cephalosporins, such as cefotaxime, ceftazidime, and ceftriaxone, and fourth-generation cephalosporins, such as cefepime. Based upon different single amino acid substitutions, currently 140 TEM type enzymes have been described [7,8,9].

A related but less common class A β-lactamase was termed SHV, since sulfhydryl reagents had a variable effect on substrate specificity. More than 60 SHV varieties are reported. SHV is found in several Enterobacteriaceae members and *P. aeruginosa.* These enzymes can hydrolyze aztreonam and third-generation cephalosporins, such as cefotaxime and ceftazidime. Clavulanic acid inhibits these enzymes, and these organisms are susceptible to cephamycins (cefoxitin) and carbapenems (imipenem). They are the predominant ESBL type in Europe and the United States and are found worldwide [7,8,9].

The class A CTX–M β-lactamases are enzymes named for their greater activity against cefotaxime and other oxyimino β-lactam substrates, such as ceftazidime, ceftriaxone, and cefepime. More than 40 CTX-M enzymes are described. A few of them are more active on ceftazidime than cefotaxime. They were found in *Salmonella enterica* serovar *Typhimurium* strains and *E. coli*, but have also been reported in other species of Enterobacteriaceae. They are the predominant ESBL type in South America. CTX- M-14, CTX-M-3, and CTX-M-2 are the most common enzymes [7,8,9]. The CTX-M family became predominant over TEM and SHV during the first decade of the 21st century [10].

The class C AmpC β-lactamases are cephalosporinases encoded on the chromosomes of several Enterobacteriaceae and a few other bacteria. They mediate resistance to cephalothin, cefazolin, cefoxitin, most penicillins, and β-lactamase inhibitor/β-lactam combinations. In many microorganisms, AmpC enzymes are inducible, and overexpression confers resistance to broad-spectrum cephalosporins, such as cefotaxime, ceftazidime, and ceftriaxone. Transmissible plasmids encoding genes for AmpC enzymes can appear in bacteria lacking or poorly expressing a chromosomal AmpC gene, such as *E. coli*, *K. pneumoniae*, and *P. mirabilis*. Carbapenems can usually be used to treat infections due to AmpC-producing bacteria, but carbapenem resistance has been reported in some organisms, since mutations can reduce influx (outer membrane porin loss) or enhance efflux (efflux pump activation). These enzymes present a very great geographic distribution and, therefore, accurate detection and characterization are important not only for epidemiological control, but also for clinical, laboratory, and infection points of view [7,8,9].

The OXA-type class D β-lactamases are carried on plasmids, except for OXA-18. They confer resistance to ampicillin and cephalothin and present hydrolytic activity against oxacillin, cloxacillin, and carbapenems. Moreover, they are susceptible to clavulanic acid (except for OXA-48 and OXA-48-like enzymes). Amino acid substitutions in OXA enzymes can confer the ESBL phenotype. The OXA-type ESBLs have been reported mainly in *P. aeruginosa*. OXA-51, OXA-23, and OXA-58 are more prevalent in *Acinetobacter* spp., due to genetic circulation and horizontal gene exchange mechanisms observed in *P. aeruginosa.* OXA-17 confers greater resistance to cefotaxime and cefepime [7,8,9].

Other plasmid-mediated ESBLs, such as PER, VEB, GES, and IBC β-lactamases, have been reported as uncommon and have been found mainly in *P. aeruginosa* [7,8,9].

The class B MBLs were found as resident chromosomally encoded enzymes in some environmental species of low pathogenic level, but since the mid-1990s, several plasmid-encoded MBLs have emerged as acquired carbapenemases in isolates of Gram-negative bacteria. The most common MBL families include the IMP, VIM, GIM, SIM, and SPM enzymes which were isolated within a variety of integron structures and, when associated with plasmids or transposons, transfer between bacteria is facilitated. IMP-1 was the first mobile MBL, which was discovered in a *P. aeruginosa* strain in Japan in 1988. IMP enzymes spread slowly to other countries, and were reported from Europe in 1997, and then in Canada and Brazil. The Verona integron-encoded MBL (VIM-1) was isolated in Verona, Italy, in 1997. It consists of 14 members which have a wide geographic distribution worldwide. VIM enzymes mainly occur in *P. aeruginosa, P. putida,* and, very rarely, in Enterobacteriaceae. The Sao Paulo MBL (SPM-1) was isolated in a *P. aeruginosa* strain in Sao Paulo, Brazil, in 1997. German imipenemase (GIM-1) was isolated in Germany in 2002. It presents 30% homology to VIM, 43% to IMPs, and 29% to SPM. Seoul imipenemase (SIM-1) was isolated in Korea in 2003. VIM and IMP are detected worldwide, with an overall trend of these two MBLs moving beyond *P. aeruginosa* and into the Enterobacteriaceae [7,8,9].

Recently, Coppi and co-authors reported the characterization of two ceftazidime–avibactam (CZA)-resistant KPC-producing *K. pneumoniae* strains, named KP-14159 and KP-8788. They were isolated from infections that occurred in a patient never treated with CZA. Whole-genome sequencing characterization showed that both isolates belonged to the same ST258 strain, which presented altered outer membrane porins (a truncated OmpK35 and an Asp137-Thr138 duplication in the L3 loop of OmpK36), and carried novel pKpQIL plasmid derivatives (pIT-14159 and pIT-8788). They harbor two copies of the Tn4401a KPC-3-encoding transposon. Plasmid pIT-8788 was a cointegrate of pIT-14159 with a ColE replicon (that was also present in KP-14159), which apparently evolved in vivo during infection. pIT-8788 was maintained at a higher copy number than pIT-14159 and, upon transfer to *E. coli* DH10B, was able to increase the CZA MIC by 32-fold. The present findings provide novel insights about the mechanisms of acquired resistance to CZA, underscoring the role that the evolution of broadly disseminated pKpQIL plasmid derivatives may have in increasing the *bla*_KPC_ gene copy number and KPC-3 expression in bacterial hosts [11].

## 3. Old and New Beta-Lactamase Inhibitors

### 3.1. Clavulanic Acid

#### 3.1.1. Chemical Structure and Mechanism of Action

Clavulanic acid is an antibiotic isolated from *Streptomyces clavuligerus* (Figure 1). Clavulanic acid contains a β-lactam ring, and differs from penicillin G and penicillin V in its second ring, which is an oxazolidine instead of a thiazolidine ring. Clavulanic acid binds strongly to β-lactamase near its active site, thereby blocking enzymatic activity and improving the antibacterial effects [2,5]. It is currently combined with amoxicillin and ticarcillin.

#### 3.1.2. Clinical Use

In the clinical setting, the β-lactamase inhibitor clavulanic acid (CA) is combined with a β-lactam, amoxicillin [12]. This molecule shows antimicrobial activity against many Gram-positive species and phenotypes, such as methicillin-sensitive *Staphylococcus aureus* (MSSA), *Staphylococcus epidermidis*, *Enterococcus faecalis, Streptococcus pyogenes,* and *Streptococcus pneumoniae*, including strains resistant to penicillin and macrolide–penicillin-resistant strains, and was active against *Streptococcus agalactiae*, penicillin-resistant *Streptococcus pneumoniae*, and Streptococci viridans. *Citrobacter freundii, Enterobacter cloacae* complex*, Hafnia alvei, Klebsiella aerogenes, Morganella morganii, Plesiomonas shigelloides, Providencia rettgeri, Providencia stuartii, Serratia marcescens, Yersinia enterocolitica, Aeromonas hydrophila, Aeromonas veronii, Aeromonas dhakensis, Aeromonas caviae,* and *Aeromonas jandaei* generally showed intrinsic resistance against amoxicillin-CA. Amoxicillin–CA had significant antimicrobial activity against many non-Enterobacteriaceae Gram-negative species, such as *H. influenzae*, *Haemophilus parainfluenzae*, and *Moraxella catarrhalis,* but was not active against *A. baumannii, Acinetobacter pittii, Acinetobacter nosocomialis, Burkholderia cepacia complex, Elizabethkingia meningoseptica, Ochrobactrum anthropi, P. aeruginosa*, and *Stenotrophomonas maltophilia.* Despite 20 years of clinical use, amoxicillin–CA maintains excellent activity against most target pathogens. Aminopenicillin/β-lactamase inhibitor combinations (e.g., amoxicillin/clavulanic acid) are well established in the therapy of a wide range of infections both in hospital and in primary care settings as a result of their broad-spectrum activity and good tolerance [12,13]. These agents are particularly suitable for the prophylaxis and treatment of polymicrobial infections. Clinical studies have demonstrated their efficacy in the treatment of diabetic foot infections, intra-abdominal infections, pulmonary infections related to aspiration, brain abscesses and pelvic inflammatory disease, and in the prophylaxis of infections after abdominal, pelvic, and head and neck surgery. Recent studies have also revealed that aminopenicillin/β-lactamase inhibitors provide therapy or prophylaxis that is more cost-effective compared to other antimicrobial agents [12,13]. The increase in consumption of amoxicillin/clavulanic acid will select for organisms resistant to both drugs [13].

### 3.2. Sulbactam

#### 3.2.1. Chemical Structure and Mechanism of Action

Sulbactam is a semi-synthetic β-lactamase inhibitor (Figure 2). The β-lactam ring of sulbactam irreversibly binds to a β-lactamase near its active site, thereby blocking these enzymes and preventing degradation of β-lactam antibiotics [2,5]. It is currently combined with ampicillin and cefoperazone.

#### 3.2.2. Clinical Use

In antimicrobial therapy, the β-lactamase inhibitor molecule sulbactam can be found in combination with ampicillin. This combination presents a wide spectrum of antibacterial activity against Gram-positive and Gram-negative microorganisms. Several studies showed the efficacy in the treatment of various bacterial infections, particularly in the ratio of 2:1. Ampicillin/sulbactam can be used in ventilator-associated pneumonia and in lower respiratory tract, gynecological/obstetric, and intra-abdominal infections. Its use is also found in acute epiglottitis and periorbital cellulitis in pediatric subjects, in diabetic foot infections, and skin and soft tissue infections. The combination is not active against *P. aeruginosa*, while it could be considered particularly active against *A. baumannii* infections. The drug is indicated as empiric therapy for a wide range of community-acquired infections in adults or children and is effective in both parenteral (ampicillin-sulbactam) and oral (as a reciprocal sultamycin prodrug) forms. In clinical trials, sultamicillin has been demonstrated to be clinically effective in adults and children against a variety of frequently encountered infections. In addition, adverse effects occur rarely, and diarrhea is reported as the most common of them. Parenteral ampicillin–sulbactam is indicated for community-acquired infections from mild to moderate severity such as intra-abdominal or gynecologic infections. In addition, it appears to be an alternative of choice for the treatment of *A. baumannii* infections caused by carbapenem-resistant strains in nosocomial sites. Collectively, ampicillin-sulbactam still represents a suitable weapon in the management of adult and pediatric infections [14,15,16].

Moreover, sulbactam can be also combined with cefoperazone, a third-generation cephalosporin, either at a fixed level of 8 mg/L sulbactam or at a fixed cefoperazone:sulbactam ratio (2:1). It exhibits better antimicrobial activities against Enterobacteriaceae, *P. aeruginosa*, and *A. baumannii*, compared to a treatment with cefoperazone alone. Cefoperazone/sulbactam, in a ratio of 1:1 or 1:2, presents higher in vitro activity against multidrug-resistant organisms (extended-spectrum β-lactamase-producing (ESBL) and AmpC-producing Enterobacteriaceae), and carbapenem-resistant *A. baumannii*, except carbapenem-resistant *P. aeruginosa.* However, a higher concentration of sulbactam may induce AmpC production. Carbapenemases (KPC- and OXA-type, and metallo-β-lactamase enzymes) cannot be inhibited by sulbactam. A few in vitro studies showed that an increased sulbactam concentration had no effect on carbapenemase-resistant *P. aeruginosa* strains, suggesting that the presence of carbapenemases or AmpC overproduction cannot be inhibited by increased sulbactam levels. Sulbactam alone presents a good activity against carbapenem-resistant *A. baumannii* strains. In conclusion, an appropriate combination of cefoperazone with sulbactam could improve clinical use and optimization in humans [17]. However, sulbactam resistance was demonstrated in *A. baumannii* strains [18].

### 3.3. Tazobactam

#### 3.3.1. Molecular Structure and Mechanism of Action

Tazobactam belongs to the class of penicillanic acids in which one of the exocyclic methyl hydrogens is replaced by a 1,2,3-triazol-1-yl group (Figure 3). It is used in the form of its sodium salt in combination with antibiotics. Tazobactam irreversibly binds to β-lactamase near its active site [2,5]. This protects other β-lactam antibiotics from β-lactamase activity. It is currently combined with piperacillin and ceftolozane.

#### 3.3.2. Clinical Use

Piperacillin/tazobactam is a combination of β-lactam/β-lactamase inhibitors with a broad spectrum of antibacterial activity against most Gram-positive and Gram-negative aerobic and anaerobic bacteria, including many β-lactamase-producing microorganisms. Clinical trials in adults showed that piperacillin/tazobactam, administered in an 8:1 ratio, is effective in subjects affected by lower respiratory tract, intra-abdominal, urinary tract, gynecologic, and skin/soft tissue infections. The combination of piperacillin/tazobactam plus an aminoglycoside is used to treat patients affected by serious nosocomial infections. In clinical trials, piperacillin/tazobactam was significantly more effective than ticarcillin/clavulanic acid in terms of clinical and microbiological outcomes in patients affected by community-acquired pneumonia. Piperacillin/tazobactam in combination with amikacin presents similar antimicrobial effects to ceftazidime plus amikacin in the treatment of ventilator-associated pneumonia, and is significantly more effective than ceftazidime plus amikacin in the empiric treatment of febrile episodes in patients with neutropenia or granulocytopenia. In other studies, the efficacy of piperacillin/tazobactam was similar to that of standard aminoglycosides and other treatment regimens in patients with intra-abdominal, skin and soft tissue, or gynecologic infections. Piperacillin/tazobactam is generally well tolerated. The most frequent adverse events are gastrointestinal symptoms (most commonly diarrhea) and skin reactions. Compared to monotherapy, the incidence of adverse events with piperacillin/tazobactam is higher when it is administered in combination with an aminoglycoside [19].

Ceftolozane/tazobactam (C/T) is a new antibiotic resulting from the combination of a novel cephalosporin, structurally similar to ceftazidime, with tazobactam. C/T is active against ESBL Enterobacteriaceae and multidrug-resistant (MDR) *P. aeruginosa*, and was recently approved for the treatment of complicated intra-abdominal infections and complicated urinary tract infections. This drug is important for clinicians in several types of infection for two reasons: (I) C/T is particularly suitable in serious suspected or documented infections due to MDR *P. aeruginosa*; (II) C/T may provide an alternative to carbapenems for the treatment of infections caused by ESBL producers, thus enabling a carbapenem-sparing strategy [20]. However, unfortunately, the emergence of resistance to ceftolozane/tazobactam has been reported recently [21].

### 3.4. Avibactam

#### 3.4.1. Chemical Structure and Mechanism of Action

Avibactam belongs to the class of azabicycloalkanes in which the amino hydrogen at position 6 is replaced by a sulfooxy group (Figure 4). It is used in the form of its sodium salt in combination with antibiotics. Avibactam is structurally different from the other clinically used β-lactamase inhibitors, since it does not contain a β-lactam core. It presents an unusual mechanism of inhibition: its inhibition proceeds in a similar fashion via the opening of the avibactam ring, but the reaction is reversible, because the deacylation leads to the regeneration of the compound and not to hydrolysis and turnover [2,5]. This mechanism highlights that avibactam is highly effective in providing protection to β-lactam antibiotics against hydrolysis caused by chromosomal and plasmid β-lactamases. It is currently combined with ceftazidime and meropenem.

#### 3.4.2. Clinical Use

Avibactam is a non-β-lactam β-lactamase inhibitor that was recently approved in the USA for use in combination with ceftazidime, a cephalosporin antibiotic drug [22]. Avibactam potentiates the antimicrobial activity of ceftazidime, which is susceptible to β-lactamases produced by Gram-negative bacteria. Pharmacokinetic parameters of both ceftazidime and avibactam are approximately linear across the dose range evaluated [4,23,24]. Renal excretion was the major pathway for the clearance of avibactam. Renal dysfunction might alter the pharmacokinetics of avibactam, and dosage adjustments should be considered in subjects affected by renal impairment. Drugs that influence renal elimination of avibactam should be avoided and/or monitored for possible impact on the pharmacokinetics of avibactam [22]. In Europe, ceftazidime–avibactam (named Zavicefta^®^) is administered for the treatment of complicated urinary tract infections, including pyelonephritis, intra-abdominal infections, hospital-acquired pneumonia, including ventilator-associated pneumonia, and other infections caused by aerobic Gram-negative microorganisms [25]. Ceftazidime–avibactam presents in vitro activity against several Gram-negative pathogens, including many extended-spectrum β-lactamase-, AmpC-, *K. pneumoniae* carbapenemase-, and OXA-48-producing Enterobacteriaceae and MDR *P. aeruginosa* isolates. Avibactam is not active against metallo-β-lactamase-producing strains and *A. baumannii* spp. The clinical efficacy of ceftazidime–avibactam was demonstrated to not be lower than that of carbapenem therapy. Ceftazidime–avibactam treatment was demonstrated to be efficacious in subjects with infections caused by ceftazidime-susceptible and non-susceptible Gram-negative bacteria. Ceftazidime–avibactam was commonly well tolerated, with a safety and tolerability profile similar to that of ceftazidime alone. Ceftazidime-avibactam represents a suitable choice for severe and difficult-to-treat infections [25]. However, the emergence of resistance to ceftazidime–avibactam combination has been recently described [21].

### 3.5. Relebactam

#### 3.5.1. Chemical Structure and Mechanism of Action

Relebactam is a diazabicyclooctane β-lactamase inhibitor (Figure 5). It differs from the structure of avibactam only for the presence of a piperidine ring attached to the carbon at position 2 of the carbamoyl group which reduces export of inhibitors from bacterial cells by producing a positive charge at physiological pH, thus enhancing antibacterial activity. Relebactam, like its predecessor avibactam, has not been observed to undergo degradation via desulfation [2,5]. It is currently combined with imipenem and cilastatin.

#### 3.5.2. Clinical Use

In 2019, the imipenem–cilastatin/relebactam (named Recarbrio™, Merck Sharp & Dohme B.V., The Netherlands) combination was approved for the treatment of adults affected by complicated urinary tract infections and complicated intra-abdominal infections [26,27]. The III clinical trials RESTORE-IMI 1 and RESTORE-IMI 2 showed that imipenem–cilastatin/relebactam (I-R) may be used to treat hospital-acquired bacterial pneumonia and ventilator-associated pneumonia caused by non-imipenem susceptible pathogens and atypical mycobacterial infections. I-R presents in vitro activity against MDR organisms including carbapenem-resistant *P. aeruginosa* and ESBL and carbapenem-resistant Enterobacteriaceae (CRE). I-R does not completely inhibit Gram-negative bacteria producing OXA-48, and inhibition was absent against pathogens with MBLs [4]. Pharmacokinetic values of relebactam and imipenem–cilastatin were linear within the dosing range evaluated. Relebactam is mainly eliminated by urinary excretion [4,23,24]. The most common adverse events were nausea (6%), diarrhea (6%), and headache (4%) [26,27]. Unfortunately, the emergence of resistance to imipenem–cilastatin/relebactam combination has been recently described [21].

### 3.6. Vaborbactam

#### 3.6.1. Chemical Structure and Mechanism of Action

Vaborbactam is a β-lactamase inhibitor constituted by a cyclic boronic acid pharmacophore (Figure 6), which has high affinity to serine β-lactamases, thanks to the formation of a stable covalent bond between the boron moiety and the active site serine residue [2,5]. It is used in combination with meropenem for intravenous administration.

#### 3.6.2. Clinical Use

Meropenem–vaborbactam was the first combination of antibiotics, a novel, cyclic, boronic acid-based β-lactamase inhibitor, and a carbapenem, to treat infections caused by carbapenem-resistant Gram-negative pathogens [28,29,30]. Vaborbactam presents biochemical, microbiologic, and pharmacologic properties adjusted for use with a carbapenem. Meropenem was defined as the best carbapenem presenting a broad-spectrum in vitro activity, safety profile, and efficacy in the treatment of severe Gram-negative infections. Meropenem–vaborbactam has demonstrated in vitro activity against KPC-producing *K. pneumoniae* (MIC_50_ values usually less than 0.06 mg/L). The approved dosing regimen of 4 g every 8 h as a 3 h infusion showed antimicrobial efficacy against the most common CRE [23,24]. In vitro and in vivo pharmacokinetic/pharmacodynamic (PK/PD) data confirmed that this dosing regimen has both bactericidal activity and prevents resistance among bacteria presenting MICs up to 8 mg/L. The optimization of PK and PD parameters contributed to the clinical success of meropenem–vaborbactam in subjects affected by complicated urinary tract infections, including acute pyelonephritis, and severe CRE infections [24]. However, the current increasing number of KPC-producing CRE highlights that pharmacovigilance and antimicrobial stewardship efforts improve patient outcomes [31]. However, the emergence of resistance to meropenem–vaborbactam combination has been recently reported [21].

### 3.7. Zidebactam

#### 3.7.1. Chemical Structure and Mechanism of Action

Zidebactam is a bicyclo-acyl hydrazide β-lactam enhancer antibiotic with a dual mode of action (Figure 7): selective and high-affinity binding to Gram-negative penicillin-binding protein 2 (PBP2) and β-lactamase inhibition [32]. It is used in combination with cefepime for parenteral administration. Zidebactam is under investigation in clinical trials.

#### 3.7.2. Clinical Use

Zidebactam is used against most *E. coli*, *K. pneumoniae*, and *Citrobacter* and *Enterobacter* spp. strains at MICs ≤2 mg/L (0.12–0.5 mg/L), but higher than 32 mg/L for species belonging to Proteeae, most *Serratia* spp., a few *E. coli*, *K. pneumoniae*, and *Enterobacter/Citrobacter* spp. strains [24,32,33]. In zidebactam-resistant isolates with class A and C enzymes, the improvement of antimicrobial activity of the cephalosporin cefepime was verified, demonstrating the β-lactamase-inhibitory activity of zidebactam. The combination cefepime/zidebactam inhibited almost all Enterobacteriaceae with AmpC, ESBL, K1, KPC, and OXA-48-like β-lactamases. The majority of AmpC, metallo-β-lactamase-producing *P. aeruginosa* isolates were susceptible to cefepime/zidebactam at MIC 8 + 8 mg/L. Zidebactam enhances the antimicrobial activity of cefepime in *S. maltophilia* strains, but on the other hand, has minimal results in *A. baumannii* isolates [24,32,33]. Karlowsky and co-authors (2020) determined the in vitro susceptibility of several clinical isolates of non-carbapenem-susceptible Enterobacteriaceae, MDR *P. aeruginosa* (also not carbapenem susceptible), *S. maltophilia*, and *Burkholderia* spp. collected worldwide. Cefepime–zidebactam inhibited 98.5% of non-carbapenem-susceptible Enterobacteriaceae at ≤8 μg/mL. Against the subset of metallo-β-lactamase (MBL)-positive Enterobacteriaceae, cefepime–zidebactam inhibited 94.9% of isolates at ≤8 μg/mL. Further, it inhibited 99.6% of MDR *P. aeruginosa* isolates at ≤32 μg/mL, including all MBL-positive isolates. Moreover, cefepime–zidebactam was active against most of Enterobacteriaceae (≥95%) and *P. aeruginosa* (99%) isolates that were not susceptible to ceftazidime–avibactam, ceftolozane–tazobactam, imipenem–relebactam, and colistin. Most of *S. maltophilia* (99%) and *Burkholderia* spp. isolates were also inhibited by cefepime–zidebactam at ≤32 μg/mL [24,34]. The activity of cefepime–zidebactam against carbapenem-resistant Gram-negative bacteria is most probably due to its β-lactam enhancer mechanism of action (binding to bacterial penicillin-binding protein 2 (PBP2)) and its universal binding stability to both serine and metallo-β-lactamases.

### 3.8. Nacubactam

#### 3.8.1. Chemical Structure and Mechanism of Action

Nacubactam is a diazabicyclooctane β-lactam enhancer antibiotic with a dual mode of action (Figure 8): selective and high-affinity binding to Gram-negative penicillin-binding protein 2 (PBP2) and β-lactamase inhibition [35]. It is used in combination with meropenem for intravenous administration. Nacubactam is under investigation in clinical trials.

#### 3.8.2. Clinical Use

Nacubactam, a novel β-lactamase inhibitor, presents two mechanisms of action: (I) inhibition of serine β-lactamases (classes A and C and some class D); (II) inhibition of penicillin-binding protein 2 [23,33,34]. The safety, tolerability, and pharmacokinetic values of nacubactam when intravenously administered have been assayed in numerous studies. Nacubactam was usually well tolerated, and the most common adverse events were headache and complications related to intravenous access. No severe cases or deaths were reported. Nacubactam pharmacokinetics were linear in an approximately dose-proportional manner among the evaluated dose ranges. Nacubactam was excreted mainly unchanged into urine. The association of nacubactam with meropenem did not significantly modify the pharmacokinetics of drugs. These results demonstrated the suitability of meropenem as one of the potential β-lactam partners for nacubactam [24,35,36]. The meropenem–nacubactam combination is effective against carbapenem-resistant *K. pneumoniae* and MBLs. Beta-lactamase inhibition by nacubactam proceeds through an alternative mechanism, similar to that for avibactam [4,37].

## 4. Discussion

The continued and often inappropriate administration of β-lactams has been the cause of the emergence and worldwide spread of extended-spectrum β-lactamases (ESBLs) and, more recently, carbapenemases, which have limited the use of all β-lactam agents, such as extended-spectrum cephalosporins (cefotaxime, ceftriaxone, and ceftazidime) and carbapenems (ertapenem, imipenem, meropenem). One of the first approaches to obviate antimicrobial resistance is to develop new stable β-lactam antibiotics, but another strategy is to discover broad-spectrum β-lactamase inhibitors in order to evade bacterial enzymatic inactivation [1,5,6,38]. Beta-lactamase inhibitors are co-administered with existing β-lactams to inhibit bacterial β-lactamases [4,5,6]. Generally, the antimicrobial spectrum of these combinations is dependent on the activity of β-lactam as well as the features of the β-lactamase inhibitor. The most administered combinations ampicillin–sulbactam, amoxicillin–clavulanate, and piperacillin–tazobactam are efficient only towards class A β-lactamases, except KPC, therefore broad-spectrum inhibitors effective against all classes of β-lactamases, including MBLs, will be desirable [2,3,4,5]. The search for more effective β-lactamase inhibitors gained new impetus in recent years by the limited discovery of antibiotics for multidrug-resistant Gram-negative bacteria. The introduction of the diazabicyclooctanones (DBOs), of which avibactam was the first compound which reached the clinic as a combination with the cephalosporin ceftazidime, was a key step [2,5]. Avibactam inhibits β-lactamases belonging to class A, including KPC, class C, and certain class D β-lactamases, mainly OXA-48. Moreover, the investigation of interactions of avibactam with β-lactamases and the several available crystal structures will be helpful to extend its inhibition activity to other enzymes, such as class D β-lactamases. The success obtained by avibactam stimulated research on others DBOs, particularly relebactam, which was more recently approved for the clinic in combination with the carbapenem imipenem [2,5]. The development of boronate-based compounds and, more recently, cyclic boronates represents a second step for β-lactamase inhibitor development. Vaborbactam is one of the most important of this new class of drugs and inhibits β-lactamases of class A, but unfortunately not OXA or MBL enzymes [2,5].

The introduction of DBOs and vaborbactam in combination with the carbapenem meropenem increased the options to treat serious infections caused by multidrug-resistant Gram-negative bacteria. However, they are not active against the β-lactamases (metallo) of class B, an aspect which remains the major challenge for inhibitor development. The worldwide efforts to design and develop MBL inhibitors have been focused on compounds that bind and/or chelate zinc ions of the active site, but the high degree of heterogeneity of these enzymes constitutes a serious problem, and no drug is yet close to the clinic [2,5,38,39,40]. Sulfur-containing compounds were traditionally studied as small molecules for MBL inhibitors, and several molecules containing free thiols, thioesters, thioketones, and thioureas were reported to possess inhibitory activity [2,3,5,38,39,41]. Numerous other compounds have been described as inhibitors of MBLs: mercaptoacetic acid and its related structural analogs, picolinic acid derivatives, dicarboxylic acids, and chelating agents, such as aspergillomarasmine A and closely related molecules [3,38,39]. More recently, important results have also been reported from investigations of inhibitors able to link to the highly conserved active site residues: esters of 3-mercapto propionic acid, the selenium compound ebselen, and colloidal bismuth sulfate [2,5,39]. Despite this advancement, the variations in amino acid residues within the active site of MBLs constitute an obstacle towards achieving broad-spectrum inhibitors.

## 5. Conclusions

In recent years, the β-lactamase field has been transformed by several factors, starting from the ubiquity of ESBL strains, then the dissemination of carbapenemases, particularly KPC enzymes, and lastly the proliferation of MBLs. Continuing investigations and development of β-lactamase inhibitors are needed, especially because the continuous emergence of bacterial strains which can avoid new inhibitor combinations demonstrates that a lot of work remains to be done.

## Figures and Tables

**Figure 1 antibiotics-10-00995-f001:**
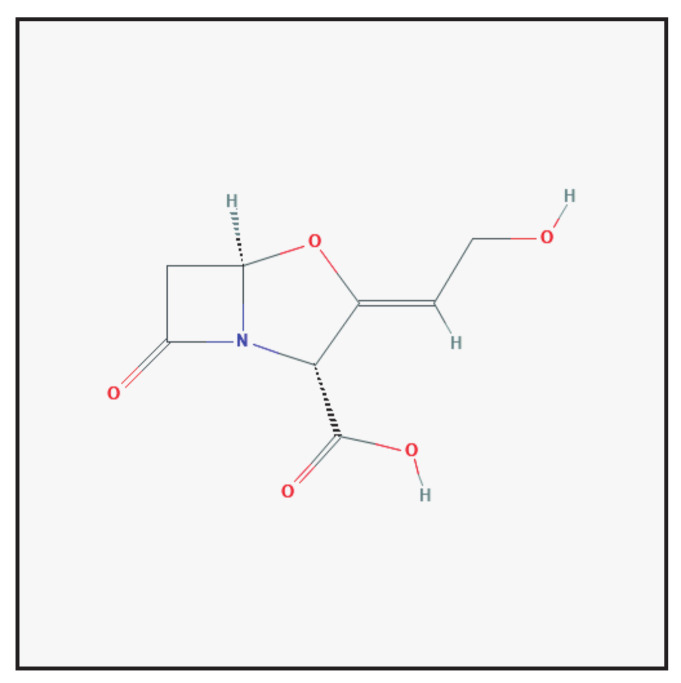
Chemical structure depiction of clavulanic acid (https://pubchem.ncbi.nlm.nih.gov (accessed on 1 May 2021)).

**Figure 2 antibiotics-10-00995-f002:**
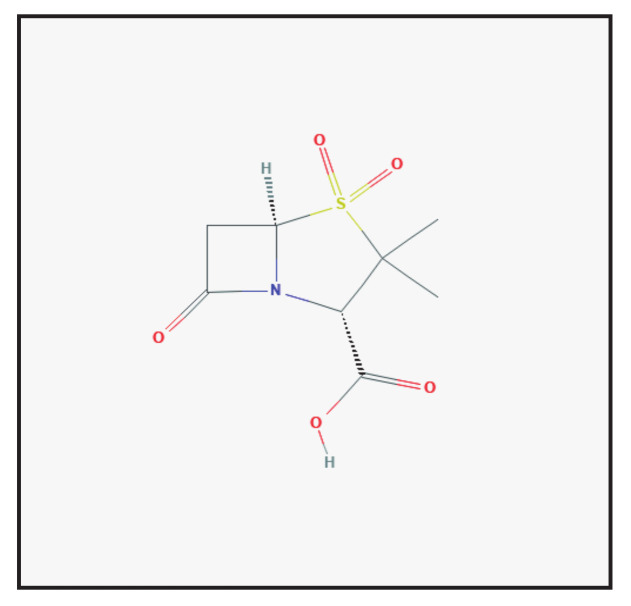
Chemical structure depiction of sulbactam (https://pubchem.ncbi.nlm.nih.gov (accessed on 1 May 2021)).

**Figure 3 antibiotics-10-00995-f003:**
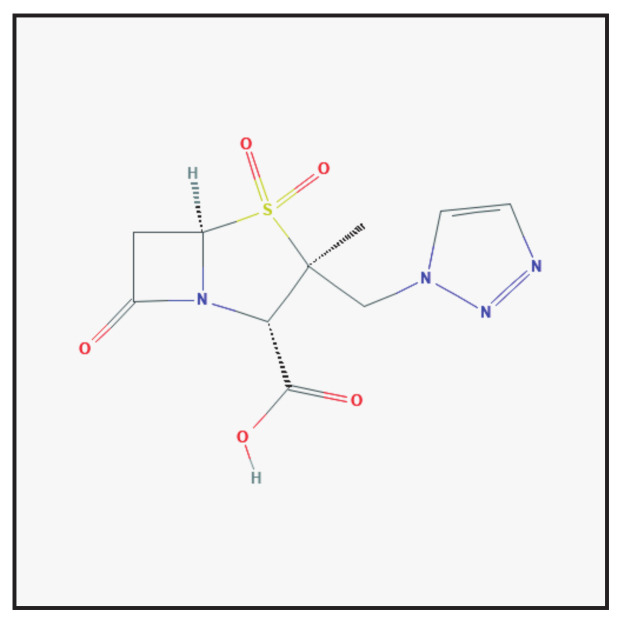
Chemical structure depiction of tazobactam (https://pubchem.ncbi.nlm.nih.gov (accessed on 1 May 2021)).

**Figure 4 antibiotics-10-00995-f004:**
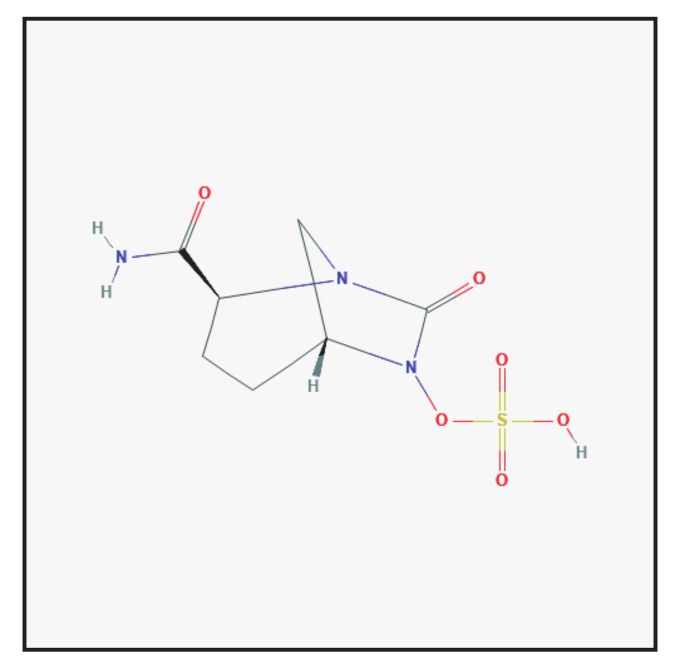
Chemical structure depiction of avibactam (https://pubchem.ncbi.nlm.nih.gov (accessed on 1 May 2021)).

**Figure 5 antibiotics-10-00995-f005:**
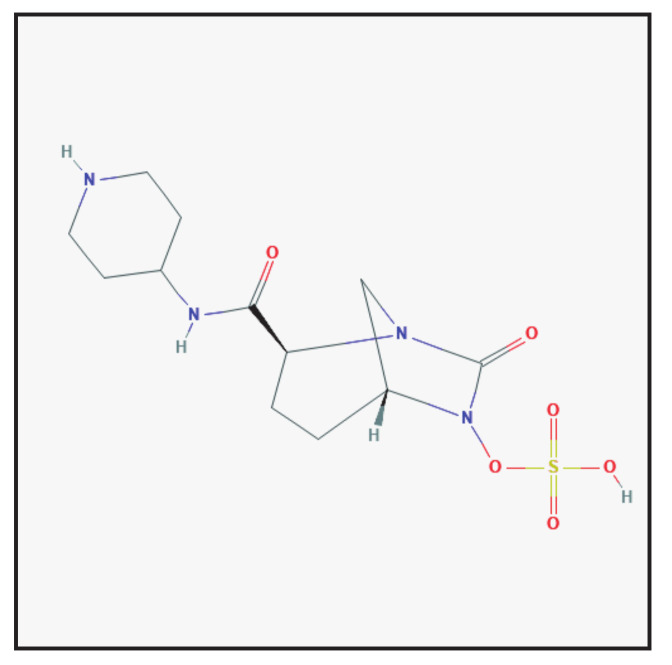
Chemical structure depiction of relebactam (https://pubchem.ncbi.nlm.nih.gov (accessed on 1 May 2021)).

**Figure 6 antibiotics-10-00995-f006:**
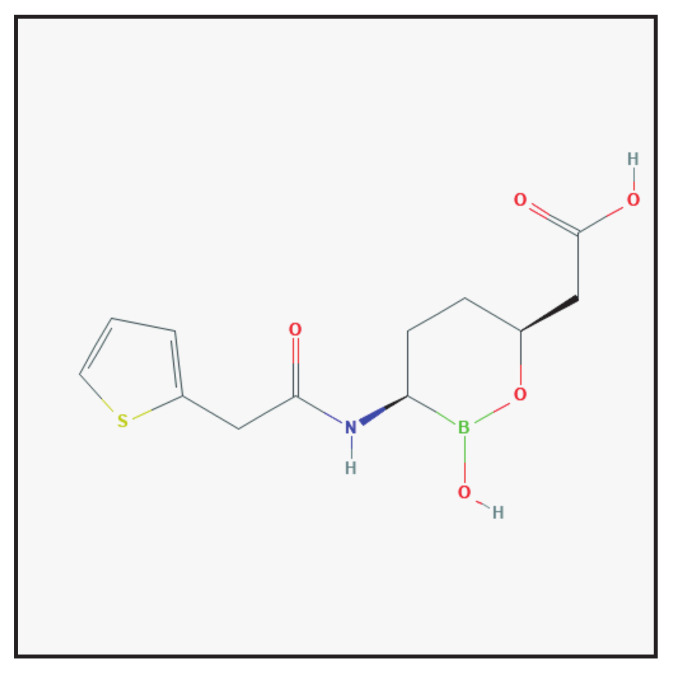
Chemical structure depiction of vaborbactam (https://pubchem.ncbi.nlm.nih.gov (accessed on 1 May 2021)).

**Figure 7 antibiotics-10-00995-f007:**
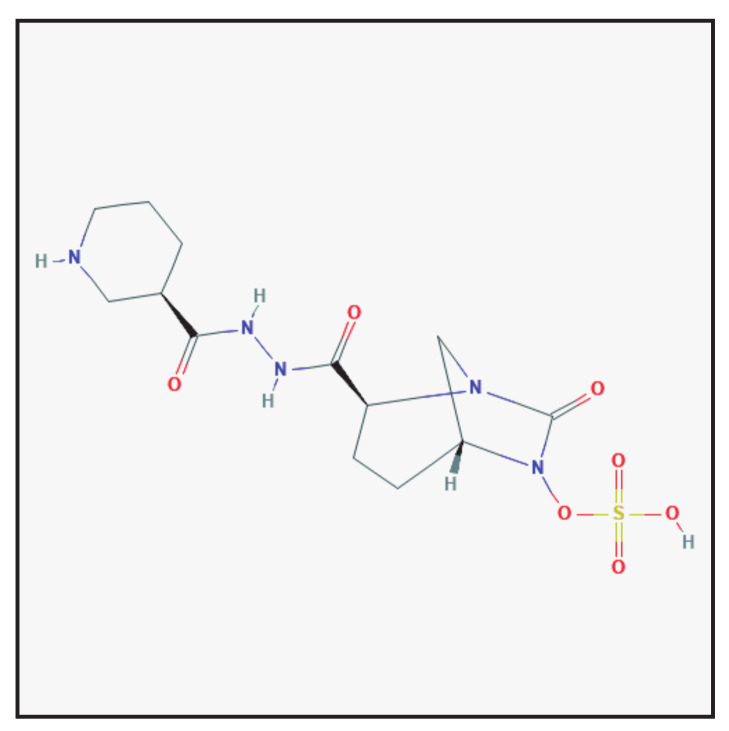
Chemical structure depiction of zidebactam (https://pubchem.ncbi.nlm.nih.gov (accessed on 1 May 2021)).

**Figure 8 antibiotics-10-00995-f008:**
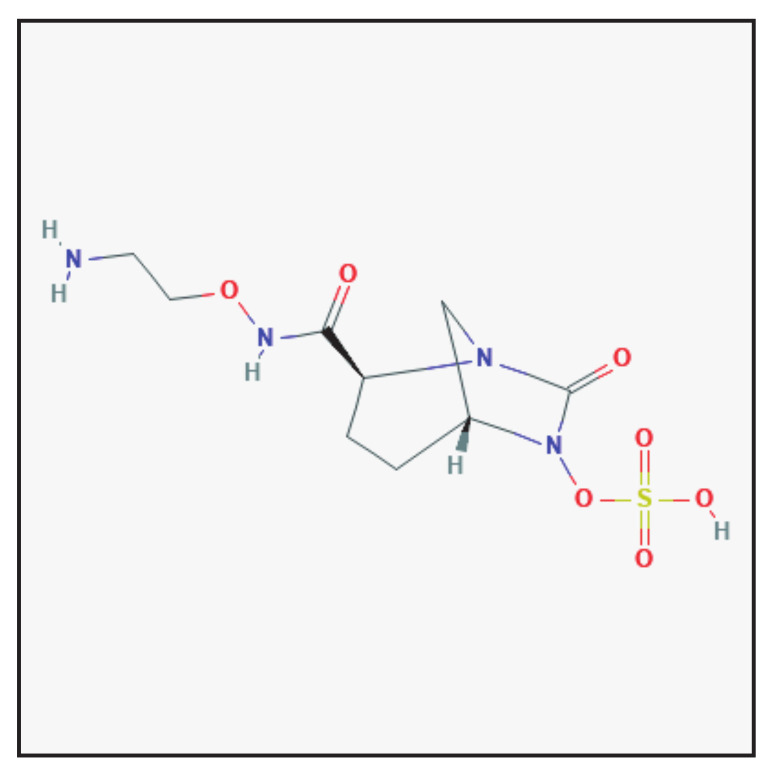
Chemical structure depiction of nacubactam (https://pubchem.ncbi.nlm.nih.gov (accessed on 1 May 2021)).

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
