# Peer review of "Old and New Beta-Lactamase Inhibitors: Molecular Structure, Mechanism of Action, and Clinical Use"

_antibiotics, 2021, doi:10.3390/antibiotics10080995_

Round 1
Reviewer 1 Report
The manuscript "Old and New Beta-Lactamase Inhibitors: Molecular Structure, 2 Mechanism of Action, and Clinical Use" by Carcione et al. provides a review concerning current β-lactamase inhibitors and an update on research efforts to identify and develop new and more efficient β-lactamase inhibitors.
The manuscript is well written, organized and up to date since the most of the references cover the literature of the last 5 years.
The manuscript deserve publication after minor revision. Few comments and/or suggestion could improve the quality of the manuscript.
- In the introduction, after the statement about the spread of resistance over the in use b-lactam antibiotics, the authors suddenly jump talking about b-lactamase inhibitors. No introduction about what b-lactamases are, and hot they act is described. To better guide the reader it would be reasonable to add a slightly detailed paragraph after line 32, introducing the common resistance mechanism adopted by bacteria to overcome the action of antibiotics, focusing on b-lactamases. Part of the paragraph 2.9 cold be moved and adapted here, thus extensevily describing the b-lactamases.
- improved the quality of the chemical structures of the b-lactamase inhibitors reported.
- for each inhibitors, in the section 'chemical structure and mechanism of action' it would prefereble to spend more words in describing the mechanism of action, also from a medicinal chemistry standpoint by reporting with pictures, if necessary, the mechanism of action and if available also reporting the inhibition constants; as an example, since the mechanism of action of avibactam is slight different from the classical b-lactam inhibitors, more information and infographic about could be a plus
- please check the abbreviations that should be reported after the first citation
- for each inhibitors, the authors, besides the clinical use, could report whether resistance phenomenon to these drug have been reported.
- since the title of the manuscript refers also to the new b-lactmase inhibitors, beside the two new drug in clinical trials, the authors could report the effort spent by the scientific community in design and evaluate new and effective compounds. The authors breafly talk about this aspect in the discussion paragraph. However, it would be better to thoroughly discuss this aspect. Several well-done reviews are reported in literature that could be at least properly cited.
- lastly, please check the manuscript for some grammatical errors and typos
Reviewer 2 Report
The article presents a well written and up-to-date synthesis on beta-lactamase inhibitors. There are mentioned also the new drugs which are under clinical trial investigation.
There are updated the data on clinical use and safety for each of the beta-lactamase inhibitors.
Author Response
Thank you for having considered our manuscript. We also thank the reviewers for their precious comments to our work.
Sincerely yours,
Jari Intra
Reviewer 3 Report
The burden of antimicrobial resistance among Gram-negative bacteria is increasing. The most relevant mechanism of resistance in Gram-negative bacteria is the presence of β-lactamases. This work provides a comprehensive overview of b-lactam inhibitors that are currently in use, as well as a look ahead to several new compounds that are in the development pipeline.
Authors provide information about molecular structure, mechanism of action and clinical use. There is few information about PK/PD of b-lactam inhibitors and combinations of β-lactam/β-lactam inhibitors. PK/PD information could be found in specific reviews.
This work could be helpful for clinical practice.
Line 203 add : and Acinetobater spp.
Avibactam is not active 202 against metallo-β-lactamase-producing strains and Acinetobater spp..
Reviewer 4 Report
This is an excellent manuscript that provides an overview of the current use and future directions of beta lactamase inhibitors in the treatment of infections. The only suggestion that I have to improve the manuscript is to move Section 2.9 to after the Introduction. This will facilitate understanding of the basics of beta lactamase biology and assist in understanding the unique features of the current beta lactamase inhibitors.
Author Response
Thank you for having considered our manuscript. We also thank the reviewers for their precious comments to our work.
We have modified our text according to the received suggestions.
We hope that this improve version is now suitable for publication.
Sincerely yours,
Jari Intra
This is an excellent manuscript that provides an overview of the current use and future directions of beta lactamase inhibitors in the treatment of infections. The only suggestion that I have to improve the manuscript is to move Section 2.9 to after the Introduction. This will facilitate understanding of the basics of beta lactamase biology and assist in understanding the unique features of the current beta lactamase inhibitors. As suggested, we thank the reviewer and we have changed the manuscript accordingly.